# DiHiRA: Diagonal High-Rank Adaption for Large Foundation Models

## Abstract

Recent advances show that large pre-trained models excel on diverse downstream tasks, driving the popularity of parameter-efficient fine-tuning (PEFT) methods. Among them, Low-Rank Adaptation (LoRA) and its variants approximate weight updates with low-rank matrix products, achieving performance comparable to full fine-tuning while greatly reducing trainable parameters and memory cost. However, low-rank updates inherently constrain model capacity, whereas update weights derived from fully fine-tuning often produces nearly full-rank weight updates. To address this limitation, we propose DiHiRA, a simple yet highly efficient extension of LoRA. By augmenting low-rank updates with a learnable diagonal matrix, DiHiRA enables high-rank adaptation while retaining parameter efficiency. This design improves flexibility and strengthens transferability across diverse tasks. Extensive experiments demonstrate that DiHiRA achieves near full-rank adaptation with LoRA-level efficiency, consistently outperforming baselines on both computer vision and natural language processing benchmarks.

## 1 Introduction

The advent of foundation models (Brown et al., 2020; Liu et al., 2019; Devlin et al., 2019; Kirillov et al., 2023; Dosovitskiy et al., 2020) has driven transformative advances across diverse domains of artificial intelligence. Traditionally, adapting such models to downstream tasks requires fully fine-tuning, i.e., updating all model parameters independently for each task (Qiu et al., 2020; Ma et al., 2024). While effective, this paradigm incurs prohibitive computational costs as model size grows, and demands substantial storage to maintain separate fine-tuned variants.

To address these limitations, recent efforts have turned to parameter-efficient fine-tuning (PEFT) (Si et al., 2025; Hu et al., 2022; Brown et al., 2020; Li & Liang, 2021; Si et al., 2024b), which seeks to retain the performance of full fine-tuning while substantially reducing the number of trainable parameters. Instead of modifying the base architecture, PEFT methods introduce lightweight trainable modules or selectively update a small subset of parameters, thereby alleviating both computational and storage burdens. Among these approaches, Low-Rank Adaptation (LoRA) (Hu et al., 2022) has emerged as a representative method, approximating weight updates via the product of two low-rank matrices. LoRA achieves strong adaptation with minimal parameter overhead, and its updates can be seamlessly merged with pre-trained weights during inference, eliminating additional runtime cost. This efficiency has made LoRA the de facto choice for scaling foundation models across tasks.

However, LoRA's inherent low-rank constraint fundamentally limits the expressiveness of its updates. As illustrated in Fig. 1, fully fine-tuned models produce update matrices with high-rank structures, indicating that rich feature transformations are often necessary for optimal adaptation. Constraining updates to low rank can therefore hinder performance, particularly in tasks requiring complex representational shifts or when the rank is set excessively low. This gap highlights a central limitation of existing PEFT methods: while highly efficient, they may fall short in capturing the full diversity of adaptations required in real-world scenarios.

To bridge this gap, we propose Diagonal High-Rank Adaptation (DiHiRA), a novel PEFT framework. Inspired by matrix theory that the sum of a low-rank and a full-rank matrix can recover near full-rank capacity, DiHiRA augments LoRA's low-rank updates with a learnable diagonal matrix, providing lightweight yet high-rank expressiveness. By jointly optimizing the low-rank and diagonal components, DiHiRA preserves LoRA's efficiency while enabling substantially richer feature

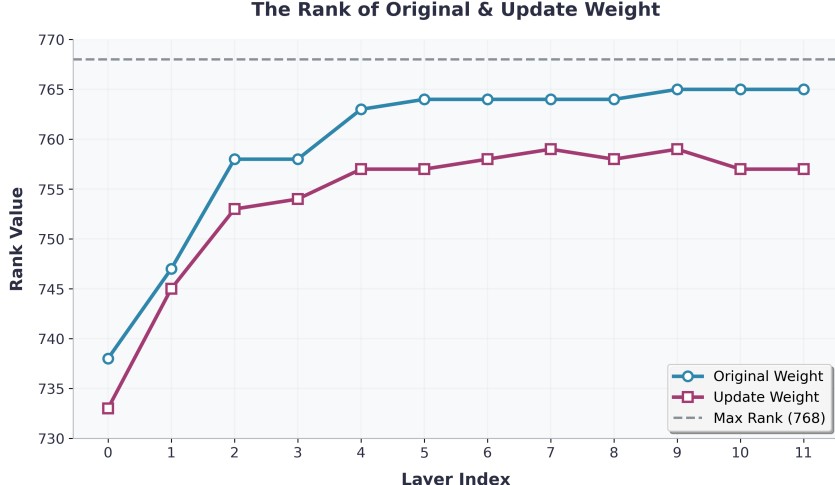

Figure 1: We count the singular values greater than 0.01 in original weight matrices $\mathbf{W}_0$ and update weight matrices $\Delta\mathbf{W}$ for the query, key, and value projections in each layer of ViT-B/16, then average the results. Here, $\Delta\mathbf{W}$ is obtained by training on the CIFAR10 (Krizhevsky et al., 2009) dataset until convergence, reaching an accuracy about 98%, which can be regarded as the optimal weight update. 768 is the maximum achievable rank.

transformations. Extensive experiments across vision and language benchmarks show that DiHiRA consistently surpasses LoRA, underscoring its effectiveness in challenging, real-world applications.

Our contributions are as follows:

- We introduce a simple-yet-effective LoRA-based PEFT method, termed DiHiRA, which pursues maintaining computational efficiency while enhancing expressiveness with just a learnable diagonal matrix.

- We provide a theoretical proof elucidating the underlying mechanism of how DiHiRA ensures high-rank updates.

- We conduct extensive empirical validation across computer vision and natural language processing tasks, justifying faster convergence and superior performance of our DiHiRA.

## 2 RELATED WORK

### 2.1 PARAMETER-EFFICIENT FINE-TUNING

As the number of parameters in models (such as Llama (Touvron et al., 2023), GPT (Brown et al., 2020), etc.) pre-trained on large-scale datasets continues to grow, the cost of full fine-tuning becomes prohibitively high. In this context, parameter-efficient fine-tuning (PEFT) methods aim to adapt these large models to various downstream tasks by fine-tuning only a portion of their parameters. Most PEFT methods are designed for the Transformer (Vaswani et al., 2017) architecture. Among these, *Prompt-based* methods (Li & Liang, 2021; Zhou et al., 2022; Jia et al., 2022; Lu et al., 2022) insert additional soft trainable tokens into the input sequence and fine-tune these tokens. However, extra tokens introduce additional computational overhead to the Transformer's attention mechanism. *Adapter-based* methods (Houlsby et al., 2019; Chen et al., 2022; Jie & Deng, 2023; Rebuffi et al., 2017; Si et al., 2024c) typically insert task-specific modules into the network to adapt to downstream tasks, but these modules also incur extra inference latency. To avoid imposing additional burdens on network inference, some methods attempt to fine-tune only the bias vectors (Zaken et al., 2021), although such approaches often yield less robust performance.

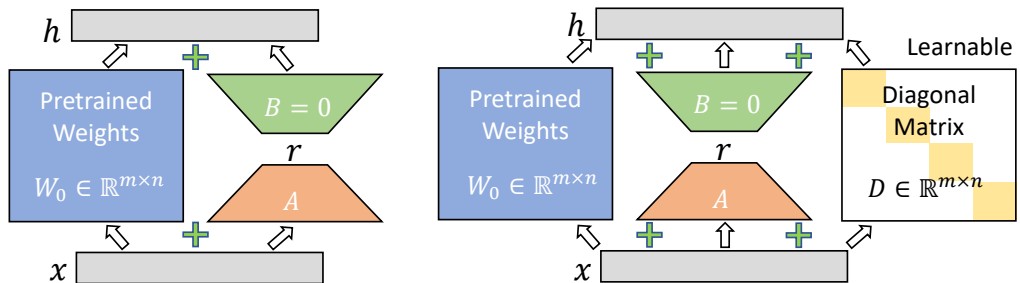

Figure 2: An overview of our proposed DiHiRA, which primarily adds a learnable diagonal matrix **D** to the LoRA framework. DiHiRA effectively harnesses the enhanced expressive capacity provided by high-rank updates, thereby achieving superior performance.

## 2.2 LOW RANK ADAPTATION

An alternative approach is offered by LoRA (Hu et al., 2022), which updates only a low-rank decomposition of the weight matrices. Specifically, instead of directly training the full weight update $\Delta \mathbf{W} \in \mathbb{R}^{m \times n}$, LoRA decomposes it into two much smaller matrices $\mathbf{A} \in \mathbb{R}^{r \times n}$ and $\mathbf{B} \in \mathbb{R}^{m \times r}$, where $r \ll \min(m, n)$ denotes the chosen rank. This factorization yields:

$$\Delta \mathbf{W} = \mathbf{BA}, \tag{1}$$

which greatly reduces the number of trainable parameters from $m \times n$ to $r(n + m)$. Given an input vector $\mathbf{x} \in \mathbb{R}^n$, the corresponding output $\mathbf{h} \in \mathbb{R}^m$ is then computed as:

$$\mathbf{h} = \mathbf{W}_0 \mathbf{x} + \Delta \mathbf{W} \mathbf{x} = \mathbf{W}_0 \mathbf{x} + \mathbf{BA} \mathbf{x}, \tag{2}$$

where $\mathbf{W}_0 \in \mathbb{R}^{m \times n}$ is the frozen pre-trained weight matrix.

Since the introduction of LoRA, a growing body of research has focused on analyzing and enhancing its effectiveness. For example, AdaLoRA (Zhang et al., 2023) dynamically allocates ranks to different layers, optimizing parameter efficiency. Methods such as DoRA (Liu et al., 2024) and PISSA (Meng et al., 2025) explore alternative matrix decomposition strategies and investigate the structural properties of weight updates. Meanwhile, LoRA-Dash (Si et al., 2024a) introduces the concept of task-specific directions (TSD), suggesting that the optimal updating primarily depends on these directions to adapt to different tasks.

## 2.3 LORA'S VARIANTS AND HIGH-RANK METHODS

While many studies have explored improvements to LoRA, few have examined the inherent limitations of its low-rank weight updates. MoRA (Jiang et al., 2024) highlights that the low-rank weight updating may limit the model's ability to capture richer information during fine-tuning, particularly in tasks that requiring continual memorization of domain-specific knowledge. HiRA (Huang et al., 2024) employs the Hadamard product to perform a high-rank update by pointwise multiplying the low-rank weights learned by LoRA with the original weight matrix. However, MoRA's approach necessitates a more complex structural design and forward computation for LoRA, introducing significant practical challenges in terms of implementation and deployment. Meanwhile, MoRA cannot truly achieve a high rank comparable to the original matrix, it can only perform high-rank updates on the reduced-dimensional square matrix. HiRA's effectiveness depends on the magnitude of $W_0$. If $W_0$ is too small (e.g., close to zero), the Hadamard product will also be small, leading to slow convergence. Additionally, the multiplicative operation in HiRA incurs significant computational overhead, and the method lacks theoretical guarantees regarding a lower bound on rank improvement. Therefore, we propose DiHiRA, a more convenient approach to achieve high-rank updates. Additionally, we find that weight updates in visual tasks also benefit from high-rank properties, which means DiHiRA's design is not only suitable for NLP tasks but also effectively supports CV tasks, offering a unified solution for efficient fine-tuning.

## 3 METHOD

To overcome the limitations of purely low-rank updates, we propose Diagonal High-Rank Adaptation (DiHiRA), a new parameter-efficient fine-tuning method that enables efficient high-rank updates. The framework is shown in Fig. 2. Standard LoRA decomposes the weight update into a product of two low-rank matrices. This formulation restricts the rank of the update to at most $r$, which may limit the ability of the model to capture complex task-specific adaptations—particularly in scenarios requiring richer feature transformations.

To enhance the expressive power of the updates, recall the property of the matrix rank that, for any two matrices $\mathbf{M}$ and $\mathbf{N}$ of the same dimensions, the inequality

$$\text{rank}(\mathbf{M} + \mathbf{N}) \geq |\text{rank}(\mathbf{M}) - \text{rank}(\mathbf{N})| \tag{3}$$

holds. This suggests that if we can augment LoRA's inherently low-rank update $\Delta \mathbf{W}$ by adding a full-rank matrix, the resulting $\Delta \mathbf{W}'$ will naturally have a much higher rank—full or nearly full rank—thereby enhancing its expressive power.

Guided by this principle, we introduce DiHiRA, which augments LoRA with a diagonal matrix $\mathbf{D} \in \mathbb{R}^{m \times n}$:

$$\Delta \mathbf{W}' = \mathbf{B}\mathbf{A} + \mathbf{D}. \tag{4}$$

Here, $\mathbf{D} \in \mathbb{R}^{m \times n}$ is chosen to be diagonal (or block-diagonal for non-square matrices), meaning that it contains nonzero entries only on its main diagonal. Since a diagonal matrix with all nonzero entries achieves full rank (i.e., $\text{rank}(\mathbf{D}) = \min\{m, n\}$), its inclusion effectively "lifts" the rank of $\Delta \mathbf{W}'$. More formally, since

$$\text{rank}(\Delta \mathbf{W}') \geq |\text{rank}(\mathbf{D}) - \text{rank}(\mathbf{B}\mathbf{A})|, \tag{5}$$

and under typical conditions $\text{rank}(\mathbf{D})$ is almost $\min\{m, n\}$, the overall update $\Delta \mathbf{W}'$ will be nearly full-rank, thereby enabling a more expressive transformation space.

In practice, $\mathbf{D}$ is initialized to zero, ensuring that pre-trained weights remain unperturbed at the beginning of fine-tuning. Both the low-rank factors $\mathbf{A}, \mathbf{B}$ and the diagonal matrix $\mathbf{D}$ are then jointly optimized during training, which preserves the parameter efficiency of LoRA while substantially enhancing representational capacity. During inference, $\mathbf{D}$ is merged with the pre-trained weights, incurring no additional computational overhead.

## 4 EXPERIMENTS

### 4.1 EXPERIMENT SETTINGS

We conduct comprehensive experiments in both computer vision (CV) and natural language processing (NLP) to evaluate the effectiveness of DiHiRA.

**Computer Vision.** For CV tasks, we use ViT-B/16 (Dosovitskiy et al., 2020) pretrained on ImageNet-21K (Ridnik et al., 2021). We evaluate on two benchmarks:

- VTAB-1K (Zhai et al., 2019), consisting of 19 datasets grouped into natural, specialized, and structured domains. Each dataset contains 1,000 labeled images (800 train, 200 val), following the standard protocol (Jie & Deng, 2023; Zhang et al., 2024).
- Fine-grained visual classification (FGVC): five datasets including Stanford-Cars (Krause et al., 2013), Oxford-Flowers102 (Nilsback & Zisserman, 2006), Oxford-Pets (Parkhi et al., 2012), Food-101 (Bossard et al., 2014), and FGVC-Aircraft (Maji et al., 2013). We consider 4-, 8-, and 16-shot settings, following (Zhang et al., 2024).

**Natural Language Processing.** For NLU tasks, we also evaluate on two benchmark:

- We fine-tune DeBERTaV3-base (Liu et al., 2019) on the GLUE benchmark (Wang et al., 2018), hich includes two single-sentence classification datasets, CoLA (Warstadt et al., 2019) and SST-2 (Socher et al., 2013), three similarity and paraphrase datasets, MRPC

Table 1: **Full results on the VTAB-1k benchmark**. The first block contains conventional tuning methods while the second block contains parameter-efficient tuning methods, which is the main focus in this paper. DiHiRA achieves the best overall performance, which is 0.8% higher on average than LoRA baseline.

| | # param (M) | Natural | | | | | | | Specialized | | | | Structured | | | | | | | | Average |
|---|---|---|---|---|---|---|---|---|---|---|---|---|---|---|---|---|---|---|---|---|---|
| | | Cifar100 | Caltech101 | DTD | Flower102 | Pets | SVHN | Sun397 | Camelyon | EuroSAT | Resisc45 | Retinopathy | Clevr-Count | Clevr-Dist | DMLab | KITTI-Dist | dSpr-Loc | dSpr-Ori | sNORB-Azim | sNORB-Ele | |
| Full | 85.8 | **68.9** | **87.7** | **64.3** | 87.2 | **86.9** | **87.4** | 38.8 | **79.7** | **95.7** | **84.2** | 73.9 | 56.3 | 58.6 | 41.7 | 65.5 | 57.5 | 46.7 | 25.7 | 29.1 | 68.9 |
| Linear | 0.04 | 64.4 | 85.0 | 63.2 | **97.0** | 86.3 | 36.6 | **51.0** | 78.5 | 87.5 | 68.5 | **74.0** | 34.3 | 30.6 | 33.2 | 55.4 | 12.5 | 20.0 | 9.6 | 19.2 | 57.6 |
| VPT (Jia et al., 2022) | 0.64 | **78.8** | 90.8 | 65.8 | 98.0 | 88.3 | 78.1 | 49.6 | 81.8 | 96.1 | 83.4 | 68.4 | 68.5 | 60.0 | 46.5 | 72.8 | 73.6 | 47.9 | **32.9** | 37.8 | 72.0 |
| Adapter (Houlsby et al., 2019) | 0.16 | 69.2 | 90.1 | 68.0 | 98.8 | 89.9 | 82.8 | 54.3 | 84.0 | 94.9 | 81.9 | 75.5 | 80.9 | 65.3 | 48.6 | 78.3 | 74.8 | **48.5** | 29.9 | **41.6** | 73.9 |
| LoRA$_{r=2}$ (Hu et al., 2022) | 0.26 | 68.5 | 90.2 | 69.3 | 98.9 | 90.5 | 86.0 | 53.2 | 83.4 | 95.6 | 84.8 | 74.5 | 83.0 | 66.6 | 50.2 | 81.2 | 76.5 | 47.2 | 30.3 | 40.6 | 74.5 |
| HiRA$_{r=2}$ (Huang et al., 2024) | 0.26 | 69.6 | 88.7 | **70.6** | 99.0 | 90.7 | 76.5 | **55.0** | 83.4 | 94.1 | 84.6 | 74.8 | 74.6 | 62.3 | 44.4 | 78.2 | 75.0 | 44.5 | 21.6 | 33.9 | 72.4 |
| DoRA$_{r=2}$ (Liu et al., 2024) | 0.32 | 68.7 | 90.3 | 70.1 | 99.0 | **91.0** | 86.6 | 53.4 | 84.2 | **96.4** | 84.9 | 75.4 | 83.4 | 67.7 | 50.3 | 81.6 | 79.3 | 47.5 | 30.5 | 41.0 | 75.1 |
| **DiHiRA$_{r=2}$(ours)** | 0.29 | 69.9 | **91.1** | 70.0 | 99.0 | 90.9 | 86.7 | 53.4 | **84.5** | 96.2 | **85.2** | 75.5 | **84.0** | **68.1** | **51.0** | **82.0** | **79.7** | 47.2 | 30.8 | 41.2 | **75.3** |

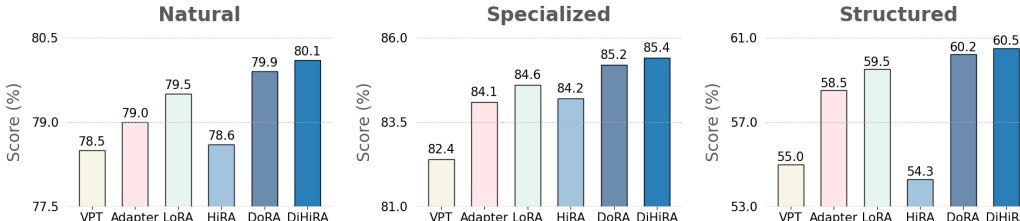

Figure 3: **Group-wise average results on VTAB-1K**. DiHiRA outperforms other PEFT methods across multiple task groups, demonstrating strong performance with only 0.29M trainable parameters. Especially in the Structured group, DiHiRA achieves a 1.0% accuracy improvement compared to LoRA, which serves as the main comparison target.

(Dolan & Brockett, 2005), QQP (Wang et al., 2018), and STS-B (Cer et al., 2017), and three natural language inference datasets, MNLI (Williams et al., 2017), QNLI (Rajpurkar et al., 2016), and RTE (Dagan et al., 2005).

- We further fine-tune LLaMA3-8B (AI@Meta, 2024) on Commonsense Reasoning (CR) benchmarks, covering eight tasks such as BoolQ, PIQA, SIQA, HellaSwag, WinoGrande, ARC-e, ARC-c, and OBQA, following the merged Commonsense170K training dataset (Si et al., 2024a; Hu et al., 2023).

**Baselines and Configurations.** We compare DiHiRA with LoRA (Hu et al., 2022), Adapters (Zhang et al., 2024), and VPT (Jia et al., 2022), alongside full fine-tuning. For CV experiments, we set rank $r = 2$ and scaling factor $s = 1.0$, applying updates to the $q, v$ projections and the FFN layers. For NLP, we use $r \in \{2, 8, 32\}$ with $s = 2$, applied to $q, k, v$ projections and FFN layers. Initialization follows prior work: $\mathbf{A}$ with Kaiming init (He et al., 2015), while $\mathbf{B}$ and $\mathbf{D}$ are set to zero. Adapter dimensions are fixed at 8, and VPT configurations follow (Jia et al., 2022).

## 4.2 MAIN RESULT

### 4.2.1 VTAB-1K BENCHMARK

Table.1 and Fig.3 shows the results on VTAB-1K. It can be observed that PEFT methods like LoRA and DiHiRA significantly outperform traditional approaches of adjusting linear layers or full fine-tuning. Compared to the main baseline LoRA, DiHiRA surpasses LoRA on most datasets, improving the average accuracy by 0.8% while only 0.03M parameters. Particularly on the Structured datasets, it achieves an average accuracy improvement of 1.0%. Additionally, compared to the Adapter with 0.16M parameters and the VPT with 0.64M parameters, DiHiRA achieves supe-

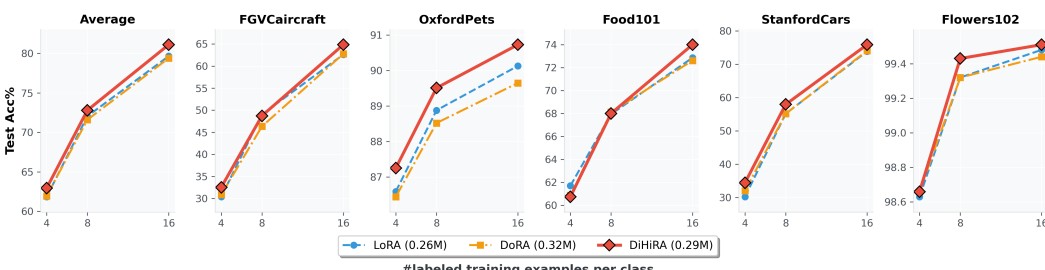

Figure 4: **Accuracy of few-shot learning on FGVC datasets.** DiHiRA outperforms other PEFT methods on average accuracy. Results are averaged over three trials with different seeds.

Table 2: **Results with DeBERTaV3 fine-tuned on GLUE**. 'FT' represents fully fine-tuning, and 'Base' represent DeBERTaV3-base. It can be observed that DiHiRA, under the settings of $r = 2$ and $r = 8$, consistently surpasses DoRA at an equivalent parameter budget while also outperforming LoRA, demonstrating its superior efficiency.

| Method | Params(%) | MNLI Acc | SST-2 Acc | CoLA Mcc | QQP Acc | QNLI Acc | RTE Acc | MRPC Acc | STS-B Corr | All Avg. |
|---|---|---|---|---|---|---|---|---|---|---|
| Base(FT) | 100% | 89.90 | 95.63 | 69.19 | 91.87 | 94.03 | 83.75 | 90.20 | 91.60 | 88.27 |
| BitFit | 0.05% | 89.37 | 94.84 | 66.96 | 88.41 | 92.24 | 78.80 | 87.75 | 91.35 | 86.21 |
| LoRA$_{r=2}$ | 0.18% | 90.03 | 93.92 | 69.15 | 90.61 | 93.37 | 87.01 | 90.19 | 90.75 | 88.13 |
| HiRA | 0.18% | 90.15 | 95.41 | 69.15 | 90.24 | 93.72 | 86.28 | 89.95 | 90.97 | 88.23 |
| DoRA | 0.22% | 90.21 | 94.38 | 69.33 | 90.84 | 93.26 | 86.94 | 90.19 | 91.34 | 88.31 |
| **DiHiRA** | 0.22% | 90.06 | 95.64 | 71.95 | 91.76 | 94.23 | 88.09 | 90.93 | 92.02 | **89.34** |
| LoRA$_{r=8}$ | 0.72% | 89.80 | 93.69 | 69.30 | 91.78 | 92.97 | 86.28 | 90.68 | 91.62 | 88.27 |
| HiRA | 0.72% | 89.94 | 95.53 | 70.19 | 91.76 | 93.96 | 86.64 | 89.71 | 91.19 | 88.62 |
| DoRA | 0.76% | 89.67 | 94.61 | 69.08 | 91.80 | 93.23 | 87.33 | 90.68 | 91.73 | 88.49 |
| **DiHiRA** | 0.76% | 89.66 | 95.87 | 69.03 | 92.12 | 94.29 | 87.36 | 92.65 | 91.74 | **89.09** |

rior performance despite using even fewer parameters. Meanwhile, compared to the LoRA-based method DoRA (Liu et al., 2024) with a similar number of additional parameters, DiHiRA demonstrates superior performance. The results demonstrate that DiHiRA can effectively enhance LoRA's performance on visual tasks, leveraging its high-rank update to improve feature representation and task-specific adaptation.

### 4.2.2 FEW-SHOT LEARNING ON FGVC

Fig.4 shows the results on FGVC tasks. In this task, we conduct experiments with three different seeds under various shot settings and average the results. The results in the figure show that DiHiRA consistently enhances the classification accuracy of LoRA and DoRA across different shot settings. This indicates that DiHiRA effectively boosts performance in FGVC tasks.

### 4.2.3 NATURAL LANGUAGE UNDERSTANDING

Table.2 shows the full results on the GLUE benchmark. Experimental results reveal that DiHiRA achieves superior performance compared to full fine-tuning while updating only 0.22% and 0.76% of the total parameters, demonstrating its remarkable parameter efficiency and effectiveness in transfer learning scenarios. Meanwhile, we observe that DiHiRA consistently outperforms LoRA on most downstream tasks. The only exceptions occur in the MNLI and COLA datasets when $r = 8$, where its performance is marginally lower than that of LoRA. Nevertheless, at the cost of only a 0.4% increase in parameter count, DiHiRA still achieves an average improvement of 0.56% over LoRA in this setting. In particular, when $r = 2$, the benefits of DiHiRA become even more significant, since it surpasses LoRA in all datasets, generating an average performance gain of 1.21%. Beyond LoRA,

Table 3: **Results on commonsense reasoning tasks**. We fine-tune LLaMA3-8B on this task. The ChatGPT used here is directly queried through the API, and its responses are obtained without any fine-tuning or additional modifications. Here, $*$ means the learning rate is scaled by $10\times$.

| Model | Method | Params(%) | BoolQ | PIQA | SIQA | HellaS. | WinoG. | ARC-e | ARC-c | OBQA | Avg. |
|---|---|---|---|---|---|---|---|---|---|---|---|
| ChatGPT | - | - | 73.1 | 85.4 | 68.5 | 78.5 | 66.1 | 89.8 | 79.9 | 74.8 | 77.0 |
| LLaMA3-8B | Fully FT | 100 | 75.3 | 89.9 | 81.5 | 95.8 | 87.6 | 91.6 | 79.3 | 87.4 | 86.1 |
| | LoRA$_{r=2}$ | 0.04 | 62.3 | 87.6 | 79.2 | 94.9 | 84.5 | 89.4 | 78.9 | 85.6 | 82.8 |
| | HiRA$^*_{r=2}$ | 0.04 | 72.1 | 87.1 | 79.6 | 94.2 | 83.7 | 90.2 | 78.3 | 84.0 | 83.7 |
| | DoRA$_{r=2}$ | 0.05 | 62.2 | 87.8 | 79.4 | 94.5 | 84.8 | 90.1 | 79.2 | 85.8 | 83.0 |
| | **DiHiRA**$_{r=2}$ | 0.05 | 72.5 | 88.1 | 80.6 | 94.5 | 84.1 | 90.2 | 78.8 | 84.8 | **84.2** |
| | LoRA$_{r=8}$ | 0.18 | 72.1 | 89.1 | 80.0 | 95.2 | 84.6 | 90.4 | 78.7 | 84.6 | 84.3 |
| | HiRA$^*_{r=8}$ | 0.18 | 73.9 | 87.5 | 80.5 | 95.3 | 83.5 | 90.0 | 80.0 | 87.2 | 84.7 |
| | DoRA$_{r=8}$ | 0.19 | 73.2 | 87.7 | 81.6 | 94.7 | 85.3 | 90.6 | 80.5 | 85.8 | 84.9 |
| | **DiHiRA**$_{r=8}$ | 0.18 | 74.8 | 88.6 | 80.5 | 94.5 | 85.3 | 90.8 | 79.8 | 86.4 | **85.1** |
| | LoRA$_{r=32}$ | 0.70 | 75.4 | 88.7 | 79.6 | 95.3 | 86.7 | 89.6 | 77.7 | 85.4 | 84.8 |
| | HiRA$^*_{r=32}$ | 0.70 | 71.1 | 88.3 | 80.4 | 95.3 | 85.4 | 90.7 | 77.7 | 87.2 | 84.5 |
| | DoRA$_{r=32}$ | 0.71 | 75.4 | 88.5 | 80.0 | 95.2 | 85.8 | 89.9 | 80.5 | 84.8 | 85.0 |
| | **DiHiRA**$_{r=32}$ | 0.71 | 75.2 | 88.5 | 80.6 | 95.3 | 87.0 | 90.4 | 78.2 | 86.2 | **85.2** |

we also compare DiHiRA with DoRA (Liu et al., 2024) under a similar parameter constraint. The experimental results demonstrate that DiHiRA exhibits superior performance.

### 4.2.4 COMMONSENSE REASONING

Table.3 shows the full results on commonsense reasoning benchmarks. In addition to fine-tuning methods, we also utilize gpt-3.5-turbo API to demonstrate the effectiveness of ChatGPT's zero-shot chain-of-thought approach as a comparison. The zero-shot CoT prompts align with the template used for collecting our commonsense fine-tuning dataset. It can be observed that, with fine-tuned parameters accounting for only 0.05% of the full model, DiHiRA achieves remarkably competitive performance. On average, it trails full fine-tuning by just 1.9% in accuracy, while outperforming LoRA at the $r = 2$ setting by 1.4%. Meanwhile, compared to the zero-shot reasoning of GPT, DiHiRA achieves an accuracy improvement of 7.2%. Across different rank settings, LoRA maintains better performance compared to DoRA with a similar parameter budget. It is evident that DiHiRA also significantly benefits the efficient fine-tuning of LLMs.

## 5 FURTHER ANALYSIS

### 5.1 RANK OF DIHIRA

We further verify that DiHiRA can indeed achieve a higher rank compared to standard LoRA. We select the ViT-B/16 model trained on the CIFAR100 dataset from the VTAB-1K benchmark and compute the rank of the update matrices $\Delta\mathbf{W}$ for the query, key, and value projections in each layer. Specifically, we quantify the effective rank by counting the number of singular values exceeding 0.01, which serves as an indicator of the number of directions that contribute significantly to the update. As shown in the left panel of Fig. 5, DiHiRA significantly increases the rank of the update matrices, nearly achieving full-rank updates.

Furthermore, we perform singular value decomposition (SVD) on the final update matrices $\Delta\mathbf{W}'$ to assess their effective rank. Although small singular values may technically contribute to the overall rank, they offer minimal expressive power. Our analysis, illustrated in the left panel of Fig. 5, demonstrates that a substantial number of singular values exceed the threshold of 0.01 at each layer. This indicates that DiHiRA can exploit a richer set of update directions than traditional LoRA, whose low-rank constraints limit its ability to capture complex feature transformations. From the visualization on the right side of Fig. 5, it can be observed that DiHiRA captures more critical

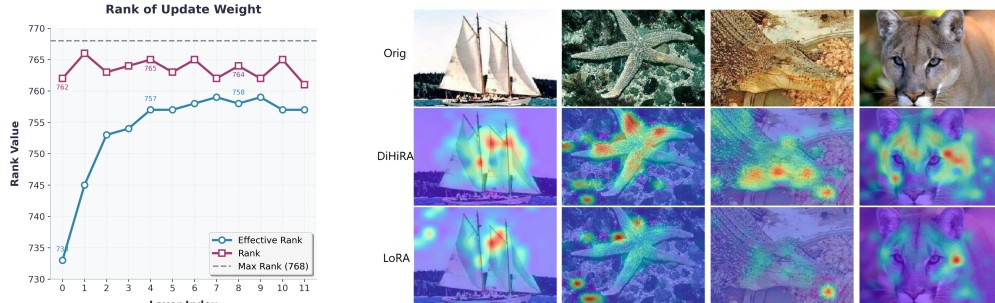

Figure 5: **Left:** We record the rank in q, v update weight matrices of each layer after applying DiHiRA and further count the number of singular values greater than 0.01, which is regarded as effective rank. **Right:** We present the attention maps of the model after fine-tuning on visual tasks, with the top, middle, and bottom corresponding to the original image, the attention map after Di-HiRA fine-tuning, and the attention map after LoRA fine-tuning, respectively.

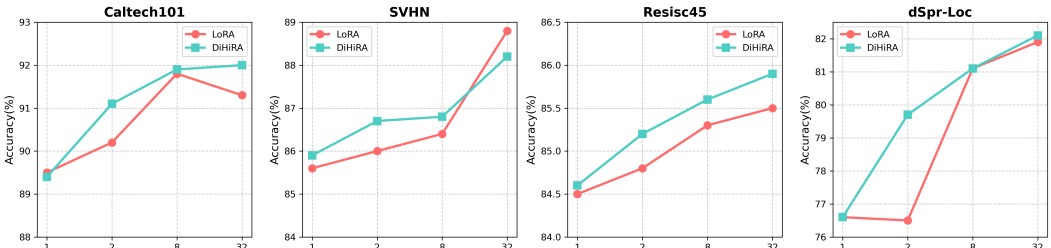

Figure 6: We further evaluate the performance of DiHiRA across various rank configurations {1, 2, 8, 32} on four distinct datasets, Caltech101, SVHN, Resisc45, and dSpr-Loc, from the VTAB-1K benchmark.

detailed features of objects. Consequently, the enhanced effective rank of DiHiRA leads to improved performance in downstream tasks.

## 5.2 PERFORMANCE ACROSS VARIED RANK

To observe the performance of DiHiRA under different rank settings, we employ ViT-B/16 as the pre-trained model and conduct tests on four distinct datasets—Caltech101, SVHN, Resisc45, and KITTI-Dist—from the VTAB-1K benchmark, using ranks of 1, 2, 8, and 32, and the results are shown in Fig. 6. Notably, we also observe a general upward trend in performance as the rank increases. When the rank is very low (e.g., 1 or 2), the improvements of DiHiRA over LoRA are more pronounced, suggesting that DiHiRA is particularly beneficial when the expressiveness of the update is severely constrained. As the rank increases to 8 and 32, the performance continues to improve but tends to saturate, indicating that beyond a certain threshold, additional rank capacity yields diminishing returns. This pattern highlights DiHiRA's ability to achieve competitive performance even under low-rank settings while maintaining a strong performance advantage in higher-rank configurations.

## 5.3 ABLATION STUDIES

We perform further ablation experiments on our DiHiRA. We attempt to use only one trainable element to control the entire identity matrix, which can be expressed by the following formula:

$$\Delta \mathbf{W} = \mathbf{B}\mathbf{A} + \alpha \mathbf{I}_{m \times n}. \tag{6}$$

Here, $\alpha$ is a learnable parameter and $\mathbf{I}_{m \times n} \in \mathbb{R}^{m \times n}$ represents a structure where the elements on the main diagonal are 1, and all other elements are 0. The initialization of $\alpha$ is set to 0. The setting of other two matrices $\mathbf{A} \in \mathbb{R}^{r \times n}, \mathbf{B} \in \mathbb{R}^{m \times r}$ are same as LoRA. We refer to this method as DiHiRA$_\alpha$.

Table 4: Results on fine-tuning ViT-B/16 with LoRA, DiHiRA$_\alpha$, and DiHiRA on Caltech101, SVHN, Resisc45, KITTI-Dist from VTAB-1K benchmark.

| Method | Params (M) | Caltech101 | SVHN | Resisc45 | KITTI-Dist |
|---|---|---|---|---|---|
| LoRA | 0.26 | 90.2 | 86.0 | 84.8 | 81.2 |
| DiHiRA$_\alpha$ | 0.26 | 90.2 | 86.5 | 84.9 | 81.0 |
| DiHiRA | 0.29 | **91.1** | **86.7** | **85.2** | **82.0** |

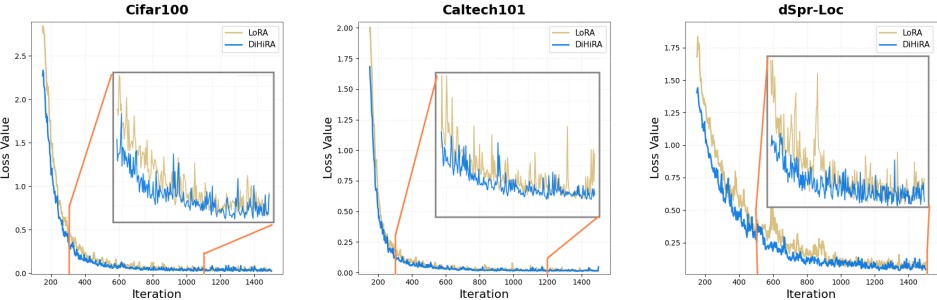

Figure 7: We record the loss curves of ViT-B/16 fine-tuned on the Cifar100, Caltech101 and dSpr-Loc datasets from VTAB-1K benchmark using LoRA and DiHiRA, respectively.

We evaluate the performance of DiHiRA and DiHiRA$_\alpha$ on the Caltech101, SVHN, Resisc45 and KITTI-Dist datasets from the VTAB-1K benchmark, with LoRA included as a baseline reference for comparative analysis, the rank of DiHiRA and DiHiRA$_\alpha$ are both 2. As shown in Table.4, DiHiRA$_\alpha$ also demonstrates the capability to enhance the performance of LoRA. However, compared to DiHiRA, DiHiRA$_\alpha$ exhibits lower accuracy on the Caltech101, SVHN, Resisc45, and KITTI-Dist datasets, trailing by 0.9%, 0.2%, 0.3% and 1.0%, respectively. DiHiRA effectively balances model performance and parameter efficiency, making it a superior choice for PEFT.

## 5.4 CONVERGENCE ANALYSIS

We further analyze the training dynamics of LoRA and DiHiRA by examining their convergence behavior. Using ViT-B/16 as the pre-trained model, we track the loss values of LoRA and DiHiRA on three datasets from the VTAB-1K benchmark, as illustrated in Fig. 7.

From the figure, it is evident that DiHiRA facilitates a significantly faster convergence compared to LoRA in the training phase. The loss values of DiHiRA decrease more rapidly, suggesting that the higher-rank update enables the model to learn meaningful representations more efficiently. This accelerated convergence can be attributed to DiHiRA's ability to leverage a richer subspace for weight adaptation, allowing for more effective gradient updates.

## 6 CONCLUSION

In this paper, we propose DiHiRA, a novel parameter-efficient fine-tuning (PEFT) method designed to address the limitations of low-rank updates in LoRA. Our approach augments LoRA's low-rank adaptation with a learnable diagonal matrix, effectively increasing the rank of the update while maintaining computational efficiency. By leveraging insights from matrix theory, we demonstrate that this simple yet effective modification significantly enhances the expressiveness of weight updates, enabling richer feature transformations. Extensive experiments across both vision and natural language tasks validate the effectiveness of DiHiRA, consistently outperforming standard LoRA. Furthermore, analysis of rank properties and convergence behavior confirms that DiHiRA provides improved adaptability and optimization efficiency. We hope our findings open new avenues for exploring high-rank PEFT strategies, contributing to the broader goal of making foundation model adaptation both efficient and expressive.

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

## A APPENDIX

We use publicly available PyTorch (Paszke et al., 2019) implementation to execute all the baseline comparisons.

### A.1 LLM USAGE

In our work, the primary roles of the large language model are to optimize the syntactic structure of text and to check for grammatical errors in the manuscript.

### A.2 EXPERIMENT DETAILS OF VISION BENCHMARK

The Images are normalized with ImageNet statistics; training uses color jitter and RandAugment, while test/validation images are resized to $256 \times 256$, center-cropped to $224 \times 224$, and normalized. All CV models are trained for 100 epochs with AdamW.

### A.3 THE HYPER-PARAMETERS OF OPTIMIZER

The hyper-parameters for VTAB-1K, FSVG, GLUE and Commonsense Reasoning tasks, as in Table. 5, 6, 7.

Table 5: Hyper-parameters for GLUE benchmark

| Dataset | MNLI | RTE | QNLI | MRPC | QQP | SST-2 | CoLA | STS-B |
|---|---|---|---|---|---|---|---|---|
| learning rate | 5e-4 | 1.2e-3 | 1.2e-3 | 1e-3 | 5e-4 | 8e-4 | 5e-4 | 2.2e-3 |
| batch size | 32 | 32 | 32 | 32 | 32 | 32 | 32 | 32 |
| epochs | 7 | 50 | 5 | 30 | 5 | 24 | 25 | 25 |

Table 6: Hyper-parameters for VTAB-1K, FSVG benchmark

| Benchmark | Batch size | Learning rate | Weight decay | Epochs | Warm-up epochs |
|---|---|---|---|---|---|
| VTAB-1K | 64 | 1e-3 | 1e-4 | 100 | 10 |
| FSVG | 64 | 1e-3 | 1e-4 | 100 | 10 |

Table 7: Hyper-parameters for commonsense reasoning task

| Method | Dropout | Batch size | Epochs | Learning rate |
|---|---|---|---|---|
| LoRA | 0.05 | 16 | 3 | 1e-4 |
| HiRA | 0.05 | 16 | 3 | 1e-3 |
| DoRA | 0.05 | 16 | 3 | 1e-4 |
| DiHiRA | 0.05 | 16 | 3 | 1e-4 |

### A.4 MORE ABLATION STUDIES

We further conduct a comparative analysis of the mechanism of the diagonal matrix $\mathbf{D}$ on the VTAB-1K benchmark datasets (Caltech101, SVHN, RESISC45, and KITTI) in the following aspects, detailed results are presented in Table. 8:

- Placement of the diagonal matrix $\mathbf{D}$: We experiment with adding the $\mathbf{D}$ matrix exclusively to the Attention (Attn) layers or the Feed-Forward Network (FFN) layers.
- Initialization of the diagonal matrix $\mathbf{D}$: We explore initializing the diagonal matrix $\mathbf{D}$ with values of 0.1, 0.5, and 1.0.

Based on further ablation results, it can be observed that incorporating the diagonal matrix in either the Attn or FFN layers leads to performance improvements, with the best results achieved when applying it to both layers simultaneously. Regarding the initialization of the diagonal matrix, initializing all elements to zero yields the optimal outcome. This aligns with the design principle of LoRA-based methods, which ensures no interference with the pre-trained model's inference at the

Table 8: Results on more ablation method

| Method | Params (M) | Caltech101 | SVHN | Resisc45 | KITTI-Dist |
|---|---|---|---|---|---|
| *Original* | | | | | |
| LoRA | 0.26 | 90.2 | 86.0 | 84.8 | 81.2 |
| DiHiRA | 0.29 | **91.1** | **86.7** | **85.2** | **82.0** |
| *Position* | | | | | |
| DiHiRA-Attn | 0.28 | 90.3 | 86.5 | 85.1 | 81.3 |
| DiHiRA-FFN | 0.28 | 90.5 | 86.3 | 85.0 | 81.5 |
| *Initialization* | | | | | |
| DiHiRA-0.1 | 0.29 | 90.1 | 86.4 | 84.9 | 81.3 |
| DiHiRA-0.5 | 0.29 | 89.6 | 86.0 | 84.9 | 80.7 |
| DiHiRA-1.0 | 0.29 | 88.2 | 85.2 | 83.0 | 78.7 |

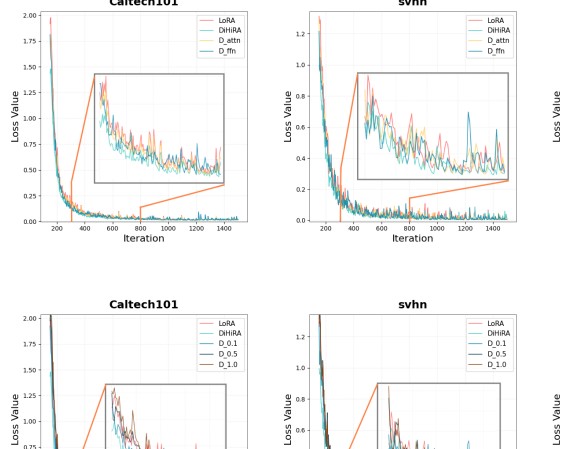
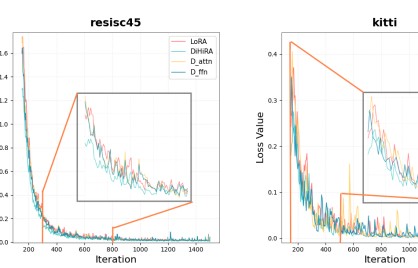
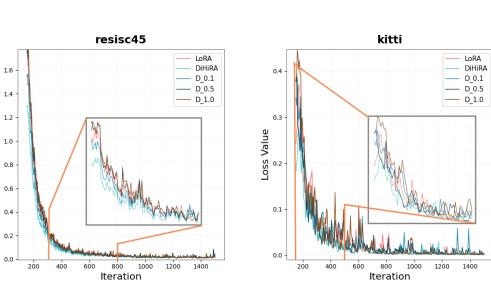

Figure 8: Training dynamics of the ablation studies: the upper figure shows the results for different placements, and the lower figure shows the results for different initialization schemes.

beginning of training. Conversely, as the initialization values increase, the growing disruption to the original pre-trained model's inference gradually degrades final performance. The whole training dynamics of these ablation studies are visualized in Fig. 8.

## A.5 COMPUTE RESOURCE STATISTICS

As shown in Table. 9, we evaluate the computational resource utilization of DiHiRA and various LoRA-based methods on the Commonsense Reasoning benchmark under a consistent experimental setup using a single A6000 GPU with $r = 32$, $r = 64$, $r = 128$ and $batchsize = 16$, the pretrained model is LLaMA3-8B.

## A.6 MORE CONVERGENCE ANALYSIS

As shown in Fig. 9, we have evaluated the training loss on the larger-scale Commonsense Reasoning benchmark using LLaMA3-8B under $r = 32$ setting. The results demonstrate that with the increased data volume and more training iterations on this benchmark, DiHiRA achieves a faster convergence speed compared to standard LoRA.

Table 9: Summary of computational efficiency

| Method | Memory (GB) | Computation Time per Epoch (seconds) |
|--------|-------------|--------------------------------------|
| $r = 32$ | | |
| LoRA | 25.3 | 32.2 |
| DoRA | 25.9 | 45.7 |
| HiRA | 25.3 | 40.5 |
| DiHiRA | 25.9 | 42.2 |
| $r = 64$ | | |
| LoRA | 25.9 | 33.0 |
| DoRA | 26.5 | 46.9 |
| HiRA | 25.9 | 44.3 |
| DiHiRA | 26.6 | 42.4 |
| $r = 128$ | | |
| LoRA | 27.7 | 33.2 |
| DoRA | 28.4 | 46.5 |
| HiRA | 27.7 | 44.4 |
| DiHiRA | 28.4 | 43.0 |

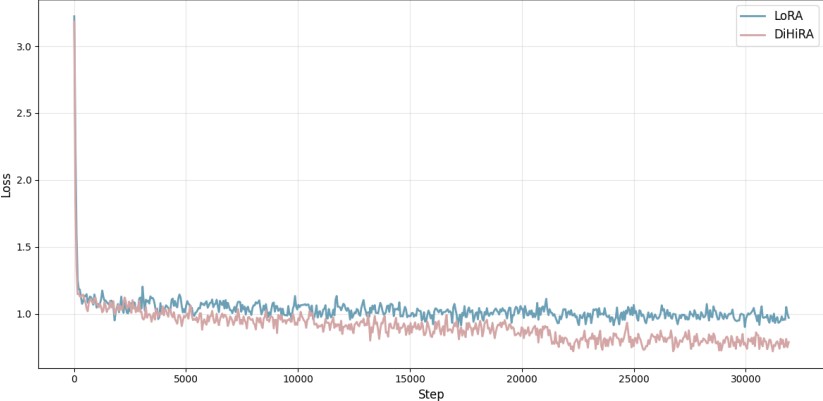

Figure 9: We record the loss curves of LLaMA3-8B fine-tuned on the Commonsense Reasoning benchmark using LoRA and DiHiRA.

