# OpenReview forum: "DiHiRA: Diagonal High-Rank Adaption for Large Foundation Models"
_ICLR.cc/2026/Conference — ICLR 2026 Conference Withdrawn Submission_

### Official Review · Reviewer_NYHE · 2025-10-30

**Soundness:** 2
**Presentation:** 3
**Contribution:** 1
**Rating:** 4
**Confidence:** 4

**Summary:**

This paper proposes DiHiRA, a novel parameter-efficient fine-tuning (PEFT) method designed to overcome the inherent low-rank constraint of Low-Rank Adaptation (LoRA). The authors argue that weight updates from full fine-tuning are high-rank, and LoRA's low-rank bottleneck can limit model expressiveness. The core idea of DiHiRA is to augment the standard LoRA update (a product of two low-rank matrices) with a learnable diagonal matrix. The paper presents empirical results across computer vision (VTAB-1K, FGVC) and natural language processing (GLUE, Commonsense Reasoning) benchmarks, demonstrating that DiHiRA consistently outperforms LoRA and other PEFT baselines with a negligible increase in trainable parameters and no additional inference cost.

**Strengths:**

DiHiRA adopts findings of previous works [1,2,3] on the low rank of LoRA limiting its representation space.

DiHiRA proposes a simple strategy to increase the rank of DoRA by adding a diagonal matrix to the DoRA factorization.

DiHiRA reports marginal accuracy gains with no notable training speed decrease compared to DoRA.

The evaluation of DiHiRA is thorough and spans multiple tasks across vision and language.

[1] Albert, Paul, et al. "RandLoRA: Full-rank parameter-efficient fine-tuning of large models." ICLR 2025.

[2] Ji, Yiping, et al. "Efficient learning with sine-activated low-rank matrices."  ICLR 2025.

**Weaknesses:**

The novelty is limited and DiHiRA can be roughly summarized as DoRA + a diagonal matrix.

Although DiHiRA adds marginal performance gains over DoRA, it also adds a small amount to extra trainable parameters. In this case it is not clear whether performance gains are (1) statistically significant (2) come from the full rank update or the added parameters.

Previous works [1] identify that the rank limitation of LoRA is mostly constraining for larger parameter budgets (LoRA rank 32 and above) yet DiHiRA is only evaluated with very small budgets which probably (as indicated in [1]) suffer more from the parameter constraint than the low rank one.

Although DiHiRA is theoretically full rank, it is not clear whether the effective rank [3] of the weight update is actually large. If the effective rank is still close to LoRA's rank then the diagonal matrix probably only marginally contributes to the improve representation space which would explain the small accuracy gains in practice.

[1] Albert, Paul, et al. "RandLoRA: Full-rank parameter-efficient fine-tuning of large models." ICLR 2025.

[3] Olivier Roy and Martin Vetterli. The effective rank: A measure of effective dimensionality. In European Signal Processing Conference, 2007

**Questions:**

Have the authors studied DiHiRA for larger ranks (32 and above) ? Have larger accuracy gains over DoRA been reported in this case ?

---

> ### Author Response · Authors · 2025-11-18
>
> Thank you for your comments! Below is the response to your question.
> > Weakness: The novelty is limited and DiHiRA can be roughly summarized as DoRA + a diagonal matrix. Although DiHiRA adds marginal performance gains over DoRA, it also adds a small amount to extra trainable parameters. In this case it is not clear whether performance gains are (1) statistically significant (2) come from the full rank update or the added parameters.
>
> **A1:** We would like to clarify that our method is not built upon DoRA. The primary motivation of DiHiRA is to efficiently transcend the low-rank constraint of LoRA and achieve a high-rank update through a simple yet effective modification—augmenting the low-rank adapter with a diagonal matrix. Theoretical underpinning for this design is provided by the matrix rank inequality: $rank(M+N)\geq|rank(M)-rank(N)|$. This guarantees that the combined update matrix maintains a higher effective rank. In terms of parameter efficiency, DoRA introduces $m$ extra parameters, whereas DiHiRA requires only $min(m, n)$ extra parameters, resulting in a strictly lower parameter count. Meanwhile, empirical results across multiple benchmarks consistently demonstrate that DiHiRA outperforms DoRA. This superior performance, achieved with greater parameter efficiency, strongly validates the effectiveness of our proposed design.
>
> > Weakness: Previous works [1] identify that the rank limitation of LoRA is mostly constraining for larger parameter budgets (LoRA rank 32 and above) yet DiHiRA is only evaluated with very small budgets which probably (as indicated in [1]) suffer more from the parameter constraint than the low rank one.
> > Q1：Have the authors studied DiHiRA for larger ranks (32 and above) ? Have larger accuracy gains over DoRA been reported in this case ?
>
> **A2:** Regarding the performance comparison, our experiments on the Commonsense Reasoning benchmark with $r=32$ demonstrate that DiHiRA outperforms DoRA. The results, presented in Table.3 of our paper, show that our method achieves superior performance.
>
> |  | BookQ | PIQA | SIQA | HellaS | WinoG | ARC-e | ARC-c | OBQA | Avg. |
> |:-:|:-:|:-:|:-:|:-:|:-:|:-:|:-:|:-:|:-:|
> | DoRA | 75.4 | 88.5 | 80.0 | 95.2 | 85.8 | 89.9 | 80.5 | 84.8 | 85.0 |
> | DiHiRA| 75.2 | 88.5 | 80.6 | 95.3 | 87.0 | 90.4 | 78.2 | 86.2 | 85.2 |
>
> > Weakness：Although DiHiRA is theoretically full rank, it is not clear whether the effective rank [3] of the weight update is actually large. If the effective rank is still close to LoRA's rank then the diagonal matrix probably only marginally contributes to the improve representation space which would explain the small accuracy gains in practice.
>
> **A3:** As illustrated in the left panel of Figure 5, our analysis of the effective rank reveals a distinct spectral characteristic: the singular values of the LoRA update matrix are primarily concentrated in the range of 0.1 to 1.0, whereas those of DiHiRA are mostly distributed between 0.01 and 0.1. In practice, since singular values do not precisely equal zero (typically on the order of 1e-6), we adopt a threshold of 0.01 to define the effective rank. Under this criterion, DiHiRA achieves a state close to full rank, demonstrating its superior capacity for parameter utilization.

---

> ### Comment · Reviewer_NYHE · 2025-11-20
>
> I thank the authors for their response and apologize for the DoRA/LoRA confusion. My intention was, as the authors correctly deducted, to suggest that DiHRA could be (roughly) summarized as LoRA + a diagonal matrix.
>
> Regarding question 3, and given that this would not require extensive experiments, could the authors report the effective rank using the formulation in [3] which is a more theoretically motivated and established manner of calculating effective rank than the approach proposed by the authors ? This would make it possible for future readers to compare with recent PEFT works [4] that also look at effective rank in PEFT (concurrently to this paper).
>
> [4] Albert, Paul, et al. "Towards Higher Effective Rank in Parameter-Efficient Fine-tuning using Khatri-Rao Product." Proceedings of the IEEE/CVF International Conference on Computer Vision. 2025.

---

> ### Author Response · Authors · 2025-11-21
>
> Thank you for your reply. Following the formula in [3] under the setting of $r=2$, we calculate the effective ranks (where the maximum rank is 768) of DiHiRA, HiRA and LoRA for ViT-B/16. The results demonstrate that DiHiRA effectively enhances the effective rank of the update matrix compared to LoRA and HiRA.
>
> | Method | Position | Efficient Rank |
> |:-:|:-:|:-:|
> | LoRA | Attn | 1.06 |
> || Mlp | 1.08 |
> | HiRA | Attn | 413.24 |
> || Mlp | 451.26 |
> | DiHiRA| Attn | 523.90 |
> || Mlp | 496.26 |

---

### Official Review · Reviewer_XC6u · 2025-10-30

**Soundness:** 2
**Presentation:** 2
**Contribution:** 2
**Rating:** 4
**Confidence:** 4

**Summary:**

The paper proposes Diagonal High-Rank Adaptation (DiHiRA), a lightweight modification to LoRA designed to overcome its low-rank bottleneck. The method augments LoRA's low-rank updates ($\Delta W = BA$) with a learnable diagonal matrix $D$, yielding $\Delta W' = BA + D$. This addition is intended to lift the rank of the updates without sacrificing efficiency.

**Strengths:**

- DiHiRA retains LoRA's mergeability (no inference overhead) and requires negligible extra storage per layer.
- Evaluation range covers both vision and language domains.

**Weaknesses:**

- The convergence analysis appears flawed. The mentioned plots do not convincingly demonstrate convergence; in fact, both DiHiRA and LoRA curves appear noisy and far from converging, suggesting neither run truly converged. The claimed faster convergence is unsupported and not convincing.
- VTAB-1k is a weak benchmark for PEFT tasks. It primarily tests data efficiency, not representational expressiveness. Moreover, the choice of rank 2 is odd; typically, LoRA and its variants are known to underperform in vision at such low ranks, which biases comparisons.
- The effective rank is ad hoc. The analysis in the paper mixes eigenvalues and singular values interchangeably; for rectangular matrices, the rank must be defined via singular values. The choice of threshold (0.01) is arbitrary and unvalidated.
- DiHiRA closely parallels recent works such as DoRA, HiRA, MoRA, etc. Yet, the paper does not convincingly argue why adding a diagonal differs materially from these or from identity-skip variants ($\alpha I$ experiment partially covers this). I would also suggest including stronger baselines.
- Statistical rigor of the improvements is thin. Most experimental results show a minimal gain, and no statistical significance test is given.
- Adding a diagonal per layer can be non-trivial for large LLMs. The paper does not include memory, activation, and wall-clock overhead analyses.
- For commonsense reasoning, HiRA uses $10\times$ higher learning rate. Are all methods equally tuned? Such discrepancies can bias results.
- The ablations seem to be limited. Numerous ablations can be made to probe the mechanism of $D$. It could include: (a) where to place $D$? it can be a selective process; (b) how to initialize $D$? (c) what if you equalize the total parameter budget compared to vanilla LoRA? (d) training stability on all the above points.

**Questions:**

- Please refer to the above weaknesses.
- "Diagonal $m\times n$" is non-standard; for $m\neq n$ this means a rectangular matrix with non-zeros on the first $\min(m,n)$ entries. Please clarify this formally in the paper.

---

> ### Author Response · Authors · 2025-11-18
>
> We thank the reviewer for your constructive comment. Our response and the corresponding revisions are as follows.
>
> > Weakness: The convergence analysis appears flawed. The mentioned plots do not convincingly demonstrate convergence; in fact, both DiHiRA and LoRA curves appear noisy and far from converging, suggesting neither run truly converged. The claimed faster convergence is unsupported and not convincing.
>
> **A1:** We thank the reviewer for pointing out the issue with the convergence plots. Our original analysis was based on an incomplete window of the training process. To address this conclusively, we have re-run the experiments and perform a full analysis of the training dynamics. The original plots have been replaced with new ones in Fig.7 of our revised version, based on complete training dynamics from steps 1 to 1500. The updated plots now convincingly demonstrate that DiHiRA reaches its stable loss level more rapidly compared to LoRA.
>
> > Weakness: VTAB-1k is a weak benchmark for PEFT tasks. It primarily tests data efficiency, not representational expressiveness. Moreover, the choice of rank 2 is odd; typically, LoRA and its variants are known to underperform in vision at such low ranks, which biases comparisons.
>
> **A2:** We have conducted additional experiments on the VTAB-1k benchmark to provide a more comprehensive comparison. We have supplemented our results with evaluations of both LoRA and DiHiRA under rank settings of $r=8$ and $r=32$.
>
> | Method | Param | 1 | 2 | 3 | 4 | 5 | 6 | 7 | 8 | 9 | 10 | 11 | 12 | 13 | 14 | 15 | 16 | 17 | 18 | 19 | Avg |
> |:-:|:-:|:-:|:-:|:-:|:-:|:-:|:-:|:-:|:-:|:-:|:-:|:-:|:-:|:-:|:-:|:-:|:-:|:-:|:-:|:-:|:-:|
> | LoRA $_{r=8}$ | 1.03M | 65.7$\pm$0.6 | 90.8$\pm$0.4 | 68.7$\pm$0.5 | 98.9$\pm$0.1 | 90.3$\pm$0.2 | 86.7$\pm$0.2 | 51.4$\pm$0.4 | 83.8$\pm$0.5 | 95.6$\pm$0.3 | 84.8$\pm$0.2 | 75.4$\pm$0.8 | 83.3$\pm$0.3 | 65.6$\pm$0.4 | 50.8$\pm$0.5 | 80.0$\pm$0.8 | 79.6$\pm$1.8 | 47.0$\pm$0.5 | 32.7$\pm$0.3 | 40.4$\pm$0.6 | 74.6 |
> | DiHiRA$_{r=8}$ | 1.07M | 67.8$\pm$0.4 | 91.5$\pm$0.4 | 69.6$\pm$0.6 | 98.9$\pm$0.1 | 90.5$\pm$0.1 | 87.1$\pm$0.1 | 51.8$\pm$0.3 | 84.3$\pm$0.5 | 95.8$\pm$0.4 | 85.3$\pm$0.3 | 75.1$\pm$0.6 | 83.7$\pm$0.5 | 65.6$\pm$0.3 | 51.2$\pm$0.5 | 80.9$\pm$0.9 | 81.2$\pm$1.4 | 48.0$\pm$0.7 | 33.3$\pm$0.2 | 40.3$\pm$0.8 | 75.1 |
> | LoRA$_{r=32}$ | 4.13M | 66.4$\pm$0.6 | 91.2$\pm$0.4 | 69.2$\pm$0.2 | 98.6$\pm$0.2 | 89.1$\pm$0.3 | 88.6$\pm$0.4 | 48.8$\pm$0.3 | 85.3$\pm$0.5 | 95.8$\pm$0.3 | 85.7$\pm$0.2 | 74.9$\pm$0.6 | 80.1$\pm$0.5 | 64.1$\pm$0.4 | 51.5$\pm$0.2 | 80.6$\pm$0.6 | 81.6$\pm$0.6 | 46.1$\pm$0.5 | 33.3$\pm$0.2 | 36.0$\pm$1.2 | 74.5 |
> | DiHiRA$_{r=32}$ | 4.17M | 67.1$\pm$0.3 | 91.8$\pm$0.3 | 69.5$\pm$0.3 | 98.7$\pm$0.2 | 89.6$\pm$0.4 | 88.8$\pm$0.6 | 49.1$\pm$0.5 | 85.9$\pm$0.9 | 95.9$\pm$0.4 | 85.8$\pm$0.3 | 75.3$\pm$0.3 | 80.3$\pm$0.9 | 64.4$\pm$0.3 | 51.5$\pm$0.4 | 80.8$\pm$0.6 | 82.2$\pm$0.3 | 47.1$\pm$0.3 | 33.3$\pm$0.9 | 36.8$\pm$0.8 | 74.8 |
>
> > Q1："Diagonal m×n" is non-standard; for m≠n this means a rectangular matrix with non-zeros on the first min(m,n) entries. Please clarify this formally in the paper.
>
> > Weakness: The effective rank is ad hoc. The analysis in the paper mixes eigenvalues and singular values interchangeably; for rectangular matrices, the rank must be defined via singular values. The choice of threshold (0.01) is arbitrary and unvalidated.
>
> **A3:** We thank the reviewer for this precise technical clarification, the term "diagonal matrix" is standardly defined for square matrices. In our method, we indeed use a rectangular matrix where only the first min(m,n) entries along the main diagonal are non-zero. And we sincerely apologize for the oversight of using eigenvalues interchangeably with singular values.
> The selection of the 0.01 threshold was based on our observation of the singular value distributions: the singular values of the LoRA update matrix are primarily distributed in the [0.1, 1.0] range, while those of DiHiRA are concentrated in the [0.01, 0.1] range. Considering that the numerical error in practical computation is approximately on the order of 1e-6, we chose 0.01 as the threshold to effectively distinguish the signal from noise for DiHiRA. We will promptly add this clarification to the manuscript.

---

> ### Author Response · Authors · 2025-11-18
>
> > Weakness: DiHiRA closely parallels recent works such as DoRA, HiRA, MoRA, etc. Yet, the paper does not convincingly argue why adding a diagonal differs materially from these or from identity-skip variants (aI experiment partially covers this). I would also suggest including stronger baselines.
>
> **A4:** Our core contribution lies in using a full, parameterized diagonal matrix to efficiently break the low-rank bottleneck of LoRA. This approach is distinct: MoRA's rank is limited by its input dimension d, and HiRA lacks a theoretical rank guarantee and slows training. While DoRA doesn't address low-rank limitations, we include it for a fair parameter-count comparison.
>
> Regarding the identity-skip ($\alpha I$) variant, our deeper ablation studies confirm that DiHiRA's non-uniform diagonal matrix provides significantly more flexible, dimension-wise adaptation than a simple global scale, which is why it performs substantially better.
>
> > Weakness: Statistical rigor of the improvements is thin. Most experimental results show a minimal gain, and no statistical significance test is given.
>
> **A5:** We agree that statistical significance testing would strengthen the results, and we will include these analyses in the final version of the paper.
>
> > Weakness: Adding a diagonal per layer can be non-trivial for large LLMs. The paper does not include memory, activation, and wall-clock overhead analyses.
>
> **A6:** We thank the reviewer for the constructive feedback. Following your suggestion, we have now included the GPU memory consumption during training and the time required per epoch on the Commonsense Reasoning benchmark for compared methods with $r=32,batchsize=16$. These efficiency metrics were measured on a single NVIDIA A6000 GPU and are presented in the table below for a direct comparison.
>
> |  | Memory Usage(GB) | 1 Epoch(second) |
> |:-:|:-:|:-:|
> | LoRA | 25.3 | 32.2 |
> | HiRA| 25.3 | 40.5 |
> | DoRA | 25.9 | 45.7 |
> | DiHiRA| 28.9 | 42.2 |
>
> > Weakness: For commonsense reasoning, HiRA uses 10× higher learning rate. Are all methods equally tuned? Such discrepancies can bias results.
>
> **A7:** Regarding the Commonsense Reasoning experiments, we would like to clarify that all methods except HiRA were evaluated under identical experimental settings. The learning rate for HiRA was specifically adjusted to 1e-3 (10x higher than the standard 1e-4) because our initial validation indicated that it failed to converge adequately at the lower rate. We attribute this need for a higher learning rate to the optimization dynamics introduced by the Hadamard product in its formulation, which appears to slow down the effective learning speed. The comparative results of HiRA at $lr=1e-4$ and $lr=1e-3$ (at $r=2$) are provided below for your reference.
>
> |  | BookQ | PIQA | SIQA | HellaS | WinoG | ARC-e | ARC-c | OBQA | Avg. |
> |:-:|:-:|:-:|:-:|:-:|:-:|:-:|:-:|:-:|:-:|
> | HiRA($lr=1e-4$) | 68.7 | 83.6 | 67.3 | 62.9 | 65.7 | 88.1 | 71.1 | 67.8 | 71.9 |
> | HiRA($lr=1e-3$) | 72.1 | 87.1 | 79.6 | 94.2 | 83.7 | 90.2 | 78.3 | 84.0 | 83.7 |
>
> > Weakness: The ablations seem to be limited. Numerous ablations can be made to probe the mechanism of D. It could include: (a) where to place ? it can be a selective process; (b) how to initialize ? (c) what if you equalize the total parameter budget compared to vanilla LoRA? (d) training stability on all the above points.
>
> **A8:** To thoroughly investigate the impact of the diagonal matrix, we have conducted extensive ablation studies as suggested. We have evaluated various insertion locations (within the FFN, within the Attn module) and different initialization schemes (0.1, 0.5, 1.0) for the D matrix. The complete training dynamics for all these configurations have been collected. The new results, now presented in Appendix A.4 of our revised manuscript. The table below shows the performance comparison of different method configurations on the Caltech101, SVHN, RESISC45, and KITTI datasets.
>
> | Method | Param | Caltech101 | SVHN | Resisc45 | Kitti |
> |:-:|:-:|:-:|:-:|:-:|:-:|
> | LoRA | 0.26 | 90.2 | 86.0 | 84.8 | 81.2 |
> | DiHiRA | 0.29 | 91.1 | 86.7 | 85.2 | 82.0 |
> |Position|
> | DiHiRA-FFN | 0.28 | 90.5 | 86.3 | 85.0 | 81.5 |
> | DiHiRA-Attn | 0.28 | 90.3 | 86.5 | 85.1 | 81.3 |
> | Init |
> | DiHiRA-0.1 | 0.29 | 90.1 | 86.4 | 84.9 | 81.3 |
> | DiHiRA-0.5 | 0.29 | 89.6 | 86.0 | 84.5 | 80.7 |
> | DiHiRA-1.0 | 0.29 | 88.2 | 85.2 | 83.0 | 78.7 |

---

> ### Comment · Reviewer_XC6u · 2025-11-27
>
> I thank the authors for their rebuttal. I have remaining concerns and follow-up questions.
>
> > The updated plots now convincingly demonstrate that DiHiRA reaches its stable loss level more rapidly compared to LoRA.
>
> Respectfully, I still do not find this claim sufficiently supported. For instance, please refer to the LoRA-GA paper [1] (see Figure 1), which provides a more convincing demonstration of both stability and performance.
>
> > VTAB-1k is a weak benchmark for PEFT tasks. It primarily tests data efficiency, not representational expressiveness.
>
> This point remains a concern for me.
>
> > we chose 0.01 as the threshold to effectively distinguish the signal from noise for DiHiRA
>
> Could the authors comment on what would happen if this threshold were selected in a more principled or data-dependent manner (e.g., via quantiles or just the min, but for the model-task)? Additionally, is the observed behavior consistent across tasks and models, as well as across different layers? The original concern of this being unvalidated remains.
>
> > Memory usage: DiHiRA [28.9 GB]
>
> This appears to add roughly +14% more memory overhead compared to LoRA. Any insight into how this overhead scales with larger models or higher ranks would benefit the paper.
>
> And +31% latency.
>
> > We agree that statistical significance testing would strengthen the results, and we will include these analyses in the final version of the paper.
>
> This would substantially strengthen the paper's claims. As I already mentioned in my original review, the reported gains do not appear to be statistically significant.
>
> Thank you again for the clarifications. These remaining questions can help further improve the final version of the paper.
>
> [1] LoRA-GA: Low-Rank Adaptation with Gradient Approximation.

---

> ### Author Response · Authors · 2025-11-28
>
> Thank you for your reply！Here is our response.
>
> > Respectfully, I still do not find this claim sufficiently supported. For instance, please refer to the LoRA-GA paper [1] (see Figure 1), which provides a more convincing demonstration of both stability and performance.
>
> **A1:** We agree that the original loss curves were noisier than those presented in LoRA-GA. This is attributable to the smaller scale of our initial plotting dataset and the relatively lower number of training iterations. To address this, we have conducted additional experiments on the Commonsense Reasoning benchmark. The new loss curves for LoRA and DiHiRA, which are presented in Appendix A6, now offer a more stable and representative visualization of the convergence behavior.
>
> > This point remains a concern for me.
>
> **A2:** The selection of the VTAB-1K benchmark is aligned with established practices in the PEFT literature. It is widely adopted by leading works in the field [1,2,3,4 etc.] as a comprehensive testbed for evaluating generalization ability across diverse visual tasks. Following this convention allows for a direct and fair comparison with state-of-the-art methods.
>
> [1] Jie, Shibo, and Zhi-Hong Deng. "Fact: Factor-tuning for lightweight adaptation on vision transformer."
>
> [2] Jie, Shibo, Haoqing Wang, and Zhi-Hong Deng. "Revisiting the parameter efficiency of adapters from the perspective of precision redundancy."
>
> [3] Jia, Menglin, et al. "Visual prompt tuning."
>
> [4] Zhang, Yuanhan, Kaiyang Zhou, and Ziwei Liu. "Neural prompt search."
>
> > Could the authors comment on what would happen if this threshold were selected in a more principled or data-dependent manner (e.g., via quantiles or just the min, but for the model-task)? Additionally, is the observed behavior consistent across tasks and models, as well as across different layers? The original concern of this being unvalidated remains.
>
> **A3:** The value of 0.01 is selected empirically based on our experiments on the VTAB-1k benchmark with the ViT-B/16 model. We observe that this value provide a consistent and meaningful distinction across different datasets and layers within this specific experimental setup. We acknowledge that this fixed threshold can be influenced by factors such as the task, layer position, and model architecture. Precisely to address the need for a more robust and universal metric, we followe the suggestion by Reviewer NYH3 and have incorporate an analysis using the Effective Rank metric[5]. Unlike a fixed threshold, effective rank provides a more stable measure. As now detailed in below, our calculations confirm that DiHiRA achieves a higher effective rank in its update matrices compared to other methods.
>
> [5] Olivier Roy and Martin Vetterli. The effective rank: A measure of effective dimensionality. In European Signal Processing Conference, 2007
>
> | Method | Position | Efficient Rank |
> |:-:|:-:|:-:|
> | LoRA | Attn | 1.06 |
> || Mlp | 1.08 |
> | HiRA | Attn | 413.24 |
> || Mlp | 451.26 |
> | DiHiRA| Attn | 523.90 |
> || Mlp | 496.26 |
>
> > Q4：Any insight into how this overhead scales with larger models or higher ranks would benefit the paper.
>
> **A4:** To provide a more comprehensive analysis, we have now extended our memory profiling experiments to include higher ranks of $r=64$ and $r=128$. And we sincerely apologize for a data transcription error in our previous report regarding the memory footprint. The correct memory usage for DiHiRA at $r=32$ is 25.9 GB, not 28.9 GB.
> | | Memory Usage(GB)| 1 Epoch(second) |
> |:-:|:-:|:-:|
> |$r=32$|||
> | LoRA | 25.3 | 32.2 |
> |HiRA | 25.3 | 40.5 |
> | DoRA | 25.9 | 45.7 |
> |DiHiRA | 25.9 | 42.2 |
> |$r=64$|||
> | LoRA | 25.9 | 33.0 |
> |HiRA | 25.9 | 44.3 |
> | DoRA | 26.5 | 46.9 |
> |DiHiRA | 26.6 | 42.4 |
> |$r=128$|||
> | LoRA | 27.7 | 33.2 |
> |HiRA | 27.7 | 44.4 |
> | DoRA | 28.4 | 46.5 |
> |DiHiRA | 28.4 | 43.0 |
>
> > Q5：As I already mentioned in my original review, the reported gains do not appear to be statistically significant.
>
> **A5:** We sincerely apologize that we have not been able to complete the requested significance tests within the rebuttal period as the experiments are still ongoing.

---

### Official Review · Reviewer_3S79 · 2025-10-30

**Soundness:** 2
**Presentation:** 3
**Contribution:** 2
**Rating:** 4
**Confidence:** 3

**Summary:**

This paper proposes DiHiRA (Diagonal High-Rank Adaptation), a simple extension of LoRA for parameter-efficient fine-tuning of large foundation models. The method augments the traditional low-rank update of LoRA with a learnable diagonal matrix, theoretically achieving near full-rank updates while maintaining efficiency. The authors provide a mathematical justification for rank improvement and conduct extensive experiments across vision (VTAB-1K, FGVC) and language benchmarks (GLUE, commonsense reasoning). Empirical results show that DiHiRA consistently outperforms LoRA and comparable PEFT methods with marginal additional parameters.

**Strengths:**

- The proposed method is conceptually straightforward and clearly described, making it easy to implement and reproduce.
- Extensive experiments across both CV and NLP domains demonstrate consistent, though moderate, performance gains over LoRA.
- The paper provides a simple but valid rank-based rationale explaining why adding a diagonal matrix can improve representational capacity.

**Weaknesses:**

(1) Limited Novelty: The core idea (adding a diagonal matrix to LoR) is conceptually incremental, offering only a small extension over existing high-rank or hybrid PEFT variants (e.g., HiRA, DoRA).

(2) Lack of Deeper Theoretical Insight: The theoretical section mainly restates rank properties rather than providing rigorous or novel analysis of optimization dynamics or generalization.

(3) Missing Efficiency Discussion: While claiming parameter efficiency, the paper does not deeply analyze training/inference time, memory usage, or deployment implications relative to existing PEFT methods.

**Questions:**

- Could the authors clarify how the diagonal matrix impacts optimization stability—does it interact with LoRA’s scaling factor or learning rate in non-trivial ways?

- Have the authors compared DiHiRA with adaptive-rank or structured variants (e.g., AdaLoRA[1], PISSA[2]) under the same parameter budget?

- Would the proposed diagonal augmentation still offer benefits when integrated with higher-rank LoRA (e.g., r ≥ 32) or larger-scale LLMs beyond 8B parameters?


**References:**

[1] Zhang, Qingru, et al. "ADAPTIVE BUDGET ALLOCATION FOR PARAMETER-EFFICIENT FINE-TUNING." In 11th International Conference on Learning Representations, ICLR 2023.

[2] Meng, Fanxu, et al. "Pissa: Principal singular values and singular vectors adaptation of large language models." In Advances in Neural Information Processing Systems 37 (2024): 121038-121072.

---

> ### Author Response · Authors · 2025-11-18
>
> We appreciate your valuable comments! We have addressed each point as follows.
>
> > Q1：Could the authors clarify how the diagonal matrix impacts optimization stability—does it interact with LoRA’s scaling factor or learning rate in non-trivial ways?
>
> **A1:** We consider that the diagonal matrix enables flexible, per-dimension adjustments, which may amplify the feature updates in certain scenarios. Nonetheless, a clear and non-trivial explanation for this underlying mechanism is still lacking.
>
> > Q2：Have the authors compared DiHiRA with adaptive-rank or structured variants (e.g., AdaLoRA[1], PISSA[2]) under the same parameter budget?
>
> **A2:** We have conducted an additional comparative experiment on the Commonsense Reasoning benchmark at rank $r=32$, explicitly evaluating against PISSA. The results demonstrate that DiHiRA achieves a clear and consistent performance advantage over PISSA.
>
> |  | BookQ | PIQA | SIQA | HellaS | WinoG | ARC-e | ARC-c | OBQA | Avg. |
> |:-:|:-:|:-:|:-:|:-:|:-:|:-:|:-:|:-:|:-:|
> | PISSA | 67.1 | 81.1 | 77.2 | 83.6 | 78.9 | 77.7 | 63.2 | 74.6 | 75.4 |
> | DiHiRA| 75.2 | 88.5 | 80.6 | 95.3 | 87.0 | 90.4 | 78.2 | 86.2 | 85.2 |
>
> > Q3：Would the proposed diagonal augmentation still offer benefits when integrated with higher-rank LoRA or larger-scale LLMs beyond 8B parameters?
>
> **A3:** Based on the comparative experiments on LLaMA3-8B with various LoRA variants at $r=32$ presented in Table.3, DiHiRA demonstrates competitive performance, which confirm that DiHiRA maintains strong performance even at higher rank settings.
>
> > Weakness：Missing Efficiency Discussion: While claiming parameter efficiency, the paper does not deeply analyze training/inference time, memory usage, or deployment implications relative to existing PEFT methods.
>
> **A4:** We have supplemented our experiments with additional measurements of GPU memory usage and the time required to complete one training epoch, conducted under the $r=32, batchsize=16$ setting, using a single A6000 GPU.
>
> |  | Memory Usage(GB) | 1 Epoch(second) |
> |:-:|:-:|:-:|
> | LoRA | 25.3 | 32.2 |
> | HiRA| 25.3 | 40.5 |
> | DoRA | 25.9 | 45.7 |
> | DiHiRA| 25.9 | 42.2 |

---

### Official Review · Reviewer_bLwM · 2025-11-01

**Soundness:** 2
**Presentation:** 2
**Contribution:** 2
**Rating:** 4
**Confidence:** 4

**Summary:**

This paper operates in the field of Parameter-Efficient Fine-Tuning (PEFT) for large foundation models. It addresses the well-known capacity limitations of Low-Rank Adaptation (LoRA), proposing that increasing the rank of the update matrix is beneficial for adaptation performance. To achieve this, the authors introduce DiHiRA, a simple extension of LoRA that augments the standard low-rank update (BA) with a learnable diagonal matrix (D). The method is evaluated on a comprehensive set of computer vision (VTAB, FGVC) and natural language (GLUE, Commonsense Reasoning) tasks. The experimental results demonstrate that DiHiRA provides notable performance gains over standard LoRA, particularly at very low-rank settings (e.g., r=2, r=4).

**Strengths:**

1. The paper tackles a relevant and important problem. Improving the expressiveness and capacity of PEFT methods while maintaining high parameter efficiency is a critical area of research for the practical deployment of foundation models.

2. The paper is well-written, clear, and easy to follow. The proposed method (DiHiRA) is simple to understand, and its implementation appears straightforward.

3. The experimental design is a significant strength. The authors have validated their method across both vision and language domains using multiple standard benchmarks, which provides a solid empirical grounding for their claims.

**Weaknesses:**

1. The primary weakness is the novelty of the core idea. The concept of moving beyond LoRA's low-rank constraint to explore high-rank or full-rank updates is not new. Several existing works have already investigated this, such as methods that merge multiple LoRA modules or re-parameterize the update in other ways [1]. Furthermore, the proposed mechanism—adding a diagonal matrix—only introduces $min(m, n)$ new parameters. While this does technically increase the rank, it is questionable how much additional representational capacity this relatively small number of parameters can truly provide for complex adaptations.

2. The paper positions itself as a high-rank adaptation method, yet its experimental evaluation is missing critical comparisons to other prominent methods in this specific space. Baselines such as MiLoRA [1] and PiSSA [2], which also aim to improve LoRA's capacity, are not compared against. This makes it difficult to contextualize the performance of DiHiRA and understand its advantages, if any, over other high-rank PEFT approaches.

3. The practical utility of the method seems questionable. The paper's own results (e.g., in Table 3 and Figure 6) clearly show that DiHiRA's performance gains are most significant at extremely low ranks ($r=2, r=4$). These settings are rarely used in practical, real-world applications, where ranks of $r=16, r=32$, or higher are common. As the rank increases to these more practical levels, the performance gap between DiHiRA and standard LoRA becomes marginal or even vanishes. This suggests the method does not offer a compelling advantage for typical use cases. Moreover, its parameter efficiency is on par with LoRA, which is less efficient than other recent PEFT families (e.g., ReFT []) that offer different performance-per-parameter trade-offs.

[1] MiLoRA: A Multi-LoRA Framework for Federated Finetuning of Large Language Models. 2024.

[2] PiSSA: Principal Singular values and singular vectors Adaptation of Large Language Models. 2023.

[3] ReFT: Representation Finetuning for Language Models. 2024.

**Questions:**

Please see my questions above.

---

> ### Author Response · Authors · 2025-11-18
>
> We are grateful to the reviewer for your thorough review and insightful comments! Our comprehensive responses are provided below.
>
> > Weakness: The primary weakness is the novelty of the core idea. The concept of moving beyond LoRA's low-rank constraint to explore high-rank or full-rank updates is not new. Several existing works have already investigated this, such as methods that merge multiple LoRA modules or re-parameterize the update in other ways [1]. Furthermore, the proposed mechanism—adding a diagonal matrix—only introduces min(m,n) new parameters. While this does technically increase the rank, it is questionable how much additional representational capacity this relatively small number of parameters can truly provide for complex adaptations.
>
> **A1:** We agree that exploring beyond strict low-rank constraints is an active research area. However, DiHiRA introduces a fundamentally different and theoretically grounded approach. Unlike prior works that rely on merging multiple LoRAs or using the Hadamard product, our method of adding a diagonal matrix to the low-rank adaptation is not only architecturally simple but also provides a theoretical guarantee for a lower bound on the rank of the update matrix. This principled design translates into compelling empirical advantages. When compared to HiRA, DiHiRA achieves superior performance across all benchmarks (as detailed in our experiments) while introducing only $min(m, n)$ additional parameters. This demonstrates a more efficient and effective use of parameters. Furthermore, the qualitative analysis in Figure 5 (right) visually confirms that DiHiRA's updates successfully capture more distinct and target-relevant features than LoRA, providing intuitive evidence for its enhanced representational capacity.
>
> > Weakness: The paper positions itself as a high-rank adaptation method, yet its experimental evaluation is missing critical comparisons to other prominent methods in this specific space. Baselines such as MiLoRA [1] and PiSSA [2], which also aim to improve LoRA's capacity, are not compared against. This makes it difficult to contextualize the performance of DiHiRA and understand its advantages, if any, over other high-rank PEFT approaches.
>
> **A2:** We have conducted additional comparative experiments on the Commonsense Reasoning benchmark at a rank of $r=32$, explicitly evaluating against MiLoRA and PISSA. The results demonstrate that DiHiRA achieves a clear and consistent performance margin over both methods. This provides further empirical evidence for the effectiveness of our proposed approach.
>
> |  | BookQ | PIQA | SIQA | HellaS | WinoG | ARC-e | ARC-c | OBQA | Avg. |
> |:-:|:-:|:-:|:-:|:-:|:-:|:-:|:-:|:-:|:-:|
> | MiLoRA | 68.8 | 86.7 | 77.2 | 92.9 | 85.6 | 86.8 | 75.5 | 81.8 | 81.9 |
> | PISSA | 67.1 | 81.1 | 77.2 | 83.6 | 78.9 | 77.7 | 63.2 | 74.6 | 75.4 |
> | DiHiRA| 75.2 | 88.5 | 80.6 | 95.3 | 87.0 | 90.4 | 78.2 | 86.2 | 85.2 |
>
> > Weakness: The practical utility of the method seems questionable. The paper's own results (e.g., in Table 3 and Figure 6) clearly show that DiHiRA's performance gains are most significant at extremely low ranks (r=2, r=4). These settings are rarely used in practical, real-world applications, where ranks of (r=16, r=32), or higher are common. As the rank increases to these more practical levels, the performance gap between DiHiRA and standard LoRA becomes marginal or even vanishes. This suggests the method does not offer a compelling advantage for typical use cases. Moreover, its parameter efficiency is on par with LoRA, which is less efficient than other recent PEFT families (e.g., ReFT []) that offer different performance-per-parameter trade-offs.
>
> **A3:** We contend that the practical significance of our method is well-justified. As shown in Table 3, taking LoRA as an example, the performance gains diminish significantly as the rank $r$ increases. For instance, the average performance improves by a mere 0.5 points when scaling $r$ from 8 to 32. Critically, our DiHiRA at $r=8$ already surpasses the performance of LoRA at $r=32$. Moreover, DiHiRA at a very low $r=2$ delivers performance on par with LoRA at $r=32$. This highlights a key practical value: DiHiRA enables the use of a much smaller $r$ to achieve—or even exceed—the performance of a larger $r$ in the original LoRA, thereby substantially saving computational resources and memory during both training and inference. Furthermore, we note that lower-rank configurations (e.g., $r=2, r=4$) often suffice for many smaller downstream tasks. Finally, we acknowledge that our work is an incremental improvement built upon the LoRA framework, and as such, there remains a performance gap compared to fundamentally different approaches like ReFT. We find this direction promising and believe exploring high-rank adaptations within the ReFT paradigm could be a valuable avenue for future work.

---

### Author Response · Authors · 2025-11-18
**Summary Comment**

Dear Area Chair,

Given the recent reversion of review scores, we understand the increased workload during this period. To assist your final assessment, we provide a summary of our **core contributions**, the **reviewer consensus on strengths**, our **revisions**, and the **updated reviewer assessments**.

### **1. Core Contributions**

We primarily investigated how to break through the intrinsic low-rank limitation of LoRA. Based on the theoretical insight that $rank(M+N)\geq|rank(M)-rank(N)|$, we propose **DiHiRA** [Fig. 2], which introduces a learnable diagonal matrix $D$  to increase the rank of the update matrix. Extensive experiments across vision and language tasks [Fig. 3-4, Table. 1-3] demonstrate that DiHiRA consistently delivers strong performance across multiple benchmarks. Subsequent analysis reveal that DiHiRA effectively enhances the rank of the update matrix [Fig. 5 left]. Visualization [Fig. 6 right] further demonstrates its ability to help vision models learn richer features. Additionally, comparison of convergence curves [Fig. 7] shows that DiHiRA aids in achieving more stable and efficient model convergence.

### **2. Reviewer Consensus on Strengths**

We are grateful for the favorable feedback provided by all reviewers:

- **Reviewer bLwM** highlighted our paper “well-written, clear, and easy to follow” and the experiment design is “a significant strength”.
- **Reviewer 3S79** found our method is "conceptually straightforward and clearly described" and “provides a simple but valid rank-based rationale explaining”.
- **Reviewer XC6u** praised our evaluation range "covers both vision and language domains".
- **Reviewer NYHE** commented our work proposes "a simple strategy" with "thorough" evaluation.

### **3. Revisions**

In accordance with the reviewers' suggestions, we have thoroughly revised the manuscript to enhance its quality. The updated PDF file incorporates substantial revisions, with the major updates outlined below:

- **More ablation study [XC6u]:** We have included further ablation experiments on the diagonal matrix $D$ in Appendix A.4. The results, presented in Table 8, offer deeper insights into its impact.
- **More ablation study [XC6u]:** We now provide the complete training dynamics of these ablation experiments, which are illustrated in Fig. 8.
- **Computational Resource Statistics [3S79 ,XC6u]:** Statistics on computational resource usage have been added in Appendix A.5, with detailed results shown in Table 9.
- **Complete Training Process [XC6u]:** The training curve in Fig. 7 has been updated to present the complete and accurate training dynamics.
- **Presentation [XC6u]:** The term "eigenvalue" has been consistently replaced with "singular value" throughout the paper, including in Section 5.1 and Fig. 5, to align with standard notation.
- **More convergence analyse [XC6u]:** We present more convergence analyse on Commonsense Reasoning benchmark in Appendix A.6.

### **4. Updated Reviewer Assessments**

Following the rebuttal phase, the status of the reviewers is as follows:

- **Reviewer XC6u:**  Several new issues have been raised:
1. Our updated complete convergence curves show insufficient improvement.
2. There remains skepticism regarding the suitability of the VTAB-1K benchmark for validating the model's representational capacity.
3. The rationale behind the chosen threshold in DiHiRA requires further clarification.
4. Memory and computational overhead across a broader range of conditions should be provided.
5. Additional significance tests need to be conducted.

We have addressed the first four points with corresponding responses and supplementary experiments. Regarding the final point, we must acknowledge with regret that due to time constraints during the rebuttal period, we were unable to complete the requested significance tests. Finally, due to the conclusion of the rebuttal phase, the reviewer did not provide further responses.
- **Reviewer NYHE:** Recognized that the previous interpretation of DiHiRA’s implementation was incorrect and further raised the question of what performance DiHiRA can achieve in terms of the effective rank. We supplemented relevant experiments; however, due to the conclusion of the rebuttal phase, the reviewer did not provide further responses.
- **Reviewers bLwM & 3S79:** We answered the questions they raised. Regrettably, as the rebuttal process was abruptly terminated, the reviewers in question were consequently unable to participate in the subsequent discussion.

We respectfully request the Area Chair to consider DiHiRA's core contributions, our revisions, and the available reviewers' feedback when drafting the meta-review.

Best regards,

Authors

---

### Note · Authors · 2026-02-15

I have read and agree with the venue's withdrawal policy on behalf of myself and my co-authors.

---

### Meta-Review · Area_Chair_8BAe · 2026-01-03

**Summary:**

After reading through the reviews, I do not think that the paper can be accepted even under normal circumstances where the reviewers could continue to adjust their scores and respond. The reviewers brought up many points that the authors could not answer; particularly the role of the diagonal matrix in optimization stability, as well as how the diagonal method differs from adding other variants. Moreover, the gains presented by the authors were marginal at best; the authors could not prove the statistical significance of their results. As such, I recommend a reject.

**Reviewer Concerns:**

The reviewers brought up many points that the authors could not answer; particularly the role of the diagonal matrix in optimization stability, as well as how the diagonal method differs from adding other variants. Moreover, the gains presented by the authors were marginal at best; the authors could not prove the statistical significance of their results. Finally, the reviewers mentioned that there is a lack of novelty. As such, I recommend a reject.

**Reviewer Scores:**

I do not think the scores will change at all.

---

### Decision · Program_Chairs · 2026-01-26

Reject